# Intranasal Mitochondrial Transplantation Restores Mitochondrial Function and Modulates Glial–Neuronal Interactions in a Genetic Parkinson’s Disease Model of *UQCRC1* Mutation

**DOI:** 10.3390/cells14151148

**Published:** 2025-07-25

**Authors:** Jui-Chih Chang, Chin-Hsien Lin, Cheng-Yi Yeh, Mei-Fang Cheng, Yi-Chieh Chen, Chi-Han Wu, Hui-Ju Chang, Chin-San Liu

**Affiliations:** 1Center of Regenerative Medicine and Tissue Repair, Institute of ATP, Changhua Christian Hospital, Changhua 50094, Taiwan; 2Department of Neurology, National Taiwan University Hospital, Taipei 10051, Taiwan; 3Department of Nuclear Medicine, National Taiwan University Hospital, Taipei 10051, Taiwan; 4Department of Nuclear Medicine, College of Medicine, National Taiwan University, Taipei 10051, Taiwan; 5Molecular Imaging Center, National Taiwan University, Taipei 106319, Taiwan; 6Vascular and Genomic Center, Institute of ATP, Changhua Christian Hospital, Changhua 50094, Taiwan; 7Department of Neurology, Changhua Christian Hospital, Changhua 50094, Taiwan; 8Graduate Institute of Integrated Medicine, College of Chinese Medicine, China Medical University, Taichung 40402, Taiwan

**Keywords:** mitochondrial transplantation, intranasal delivery, Parkinson’s disease, *UQCRC1* mutation (p.Tyr314Ser) knock-in mice, cyclosporine A, mitochondrial function, neuroprotection, inflammatory cytokines, striatal astrocytes, striatal oligodendrocytes, glial modulation

## Abstract

The intranasal delivery of exogenous mitochondria is a potential therapy for Parkinson’s disease (PD). The regulatory mechanisms and effectiveness in genetic models remains uncertain, as well as the impact of modulating the mitochondrial permeability transition pore (mPTP) in grafts. Utilizing *UQCRC1* (p.Tyr314Ser) knock-in mice, and a cellular model, this study validated the transplantation of mitochondria with or without cyclosporin A (CsA) preloading as a method to treat mitochondrial dysfunction and improve disease progression through intranasal delivery. Liver-derived mitochondria were labeled with bromodeoxyuridine (BrdU), incubated with CsA to inhibit mPTP opening, and were administered weekly via the nasal route to 6-month-old mice for six months. Both treatment groups showed significant locomotor improvements in open-field tests. PET imaging showed increased striatal tracer uptake, indicating enhanced dopamine synthesis capacity. The immunohistochemical analysis revealed increased neuron survival in the dentate gyrus, a higher number of tyrosine hydroxylase (TH)-positive neurons in the substantia nigra (SN) and striatum (ST), and a thicker granule cell layer. In SN neurons, the function of mitochondrial complex III was reinstated. Additionally, the CsA-accumulated mitochondria reduced more proinflammatory cytokine levels, yet their therapeutic effectiveness was similar to that of unmodified mitochondria. External mitochondria were detected in multiple brain areas through BrdU tracking, showing a 3.6-fold increase in the ST compared to the SN. In the ST, about 47% of TH-positive neurons incorporated exogenous mitochondria compared to 8% in the SN. Notably, GFAP-labeled striatal astrocytes (ASTs) also displayed external mitochondria, while MBP-labeled striatal oligodendrocytes (OLs) did not. On the other hand, fewer ASTs and increased OLs were noted, along with lower S100β levels, indicating reduced reactive gliosis and a more supportive environment for OLs. Intranasally, mitochondrial transplantation showed neuroprotective effects in genetic PD, validating a noninvasive therapeutic approach. This supports mitochondrial recovery and is linked to anti-inflammatory responses and glial modulation.

## 1. Introduction

Parkinson’s disease (PD) is an advanced neurodegenerative disorder characterized by the gradual loss of dopaminergic (DA) neurons in the substantia nigra (SN), leading to motor impairments and other challenging symptoms. Current therapeutic approaches encounter hurdles such as limited penetration of the blood-brain barrier and systemic side effects. Intranasal delivery has emerged as a promising strategy to overcome these challenges, offering a direct and efficient pathway to the central nervous system [1]. Previously, we demonstrated the potential of the intranasal administration of mitochondria to alleviate pathophysiological processes in both cellular and rat models of PD induced by neurotoxins [2]. Owing to the intricate nature of PD, which is influenced by environmental and genetic factors, each contributes to varying degrees of disease severity through different underlying mechanisms, achieving a recapitulation of key aspects of PD pathology. A comprehensive validation of therapeutic modalities for the intranasal transplantation of mitochondria is necessary, encompassing not only neurotoxin models, but also genetic models.

Mitochondrial ubiquinol–cytochrome c reductase core protein 1 (UQCRC1), a nuclear-encoded subunit of the mitochondrial respiratory chain complex III, plays a vital role in oxidative phosphorylation and energy production [3], with enriched expression in the mammalian brain, and specifically in the substantia nigra (SN) [4]. A missense mutation in human *UQCRC1*, c.941A > C (p.Tyr314Ser, p.Y314S), is linked to familial parkinsonism, and a reduction in neuronal UQCRC1 is associated with PD-like symptoms [5]. The PD-associated *UQCRC1* p.Y314S knock-in mouse model exhibits an age-dependent locomotor decline and loss of DA neurons [6]. Conserved UQCRC1 functions in both flies and humans, revealing a mechanism in which UQCRC1 engages cytochrome c (Cyt c), independent of reactive oxygen species (ROS), to prevent neuronal demise via caspase cascade activation [7]. These findings underscore the importance of confirming mitochondrial functional recovery in mitigating the functional pathogenicity of *UQCRC1* mutations responsible for PD. Concurrently, the therapeutic feasibility of the intranasal transplantation of mitochondria is being evaluated for its effectiveness in treating the *UQCRC1* p.Y314S knock-in transgenic model of PD.

Cyclosporin A (CsA), which is known primarily for its immunosuppressive properties, has been studied for its potential neuroprotective effects in various neurological conditions, including stroke [8], traumatic brain injury [9], PD [10,11], and neurodegenerative diseases [12]. One of the mechanisms through which CsA may exert neuroprotection is its ability to inhibit the opening of the mitochondrial permeability transition pore (mPTP), which is achieved by forming a complex with cyclophilin D (CypD) [13], leading to protection against mitochondrial dysfunction and subsequent neuronal damage [11]. In addition, the CsA pretreatment of oocytes not only prevents mitochondrial membrane potential loss and ATP reduction against oxidative stress [14], but also maintains mitochondrial function in oocytes during vitrification [15], thus increasing the developmental potential of oocytes.

Expanding on our previous findings, where we prevented calcium-induced mitochondrial swelling in CsA-treated isolated mitochondria (Mito-CsA) [16], we propose that Mito-CsA may circumvent the inevitable mitochondrial damage that occurs during the isolation process and thereby enhance the effectiveness of mitochondrial transplantation. Thus, this study aimed to assess the effectiveness of intranasal mitochondrial transplantation in treating a *UQCRC1* mutant model of PD, and to compare the treatment outcomes between mitochondria pretreated with CsA and those without pretreatment.

## 2. Materials and Methods

### 2.1. Animal Study

All animal experimental procedures adhered to the guidelines set by the Association for Assessment and Accreditation of Laboratory Animal Care (AAALAC), and received approval from the Animal Experiments and Ethics Committee of Changhua Christian Hospital (approval number CCH-AE-108-009).

### 2.2. Mitochondrial Labeling, Isolation, and Cyclosporin A Pretreatment

Mouse liver mitochondria were extracted after the in vivo labeling of mtDNA with 5-bromodeoxyuridine (BrdU) (Sigma-Aldrich, Burlington, MA, USA) for tracking postdelivery. Incorporated BrdU was detected using an anti-BrdU antibody, of which the binding affinity has been previously validated at approximately 50% [17]. Briefly, after the mice were euthanized 36 h after a single intraperitoneal (ip) injection of BrdU (0.5 mg/g), the harvested liver tissues were homogenized in ice-cold buffer (210 mM mannitol, 70 mM sucrose, 5 mM Tris-HCl, and 1 mM EDTA, pH 7.4) via a Precellys homogenizer (Bertin Technologies, Rockville, MD, USA). Mitochondria were isolated through filtration and centrifugation as previously described [18], and then suspended in a MiR05 respiration buffer (20 mM HEPES, 10 mM KH_2_PO_4_, 110 mM sucrose, 20 mM taurine, 60 mM K-lactobionate, 0.5 mM EGTA, 3 mM MgCl_2_, and 1 g/L fatty acid-free BSA) (Oroboros Instruments, Innsbruck, Austria) for CsA treatment. Mitochondrial protein was quantified via a bicinchoninic acid (BCA) assay (Pierce Biotechnology, Rockford, IL, USA), with bovine serum albumin (BSA) used as the standard.

Isolated mitochondria were treated with 5 µM CsA (Sigma-Aldrich) at a concentration of 5 µg/µL for 10 min at room temperature and washed three times by centrifugation (10,000 rpm for 3 min) at 4 °C to remove nonbinding CsA. This dosage of CsA was previously confirmed in our research to shield isolated mitochondria from calcium-induced swelling [16]. After the supernatant was removed, the mitochondrial pellets were suspended in a MiR05 buffer, and the CsA concentration was quantified via spectrophotometry to detect the absorbance at 210 nm (Appendix A). The CsA concentration absorbed by isolated mitochondria was determined via a standard curve of known concentrations (Appendix A). Mito-CsA kept on ice was immediately used.

### 2.3. Generation of Mutant UQCRC1 Knock-In Mice and Nasal Administration

The *UQCRC1* mutation (p.Y314S) knock-in mice generated via CRISPR/Cas9-based gene manipulation were obtained from the laboratory of Dr. Chin-Hsien Lin [6], and were backcrossed with WT C57BL/6J mice (BioLASCO, Taipei, Taiwan) for at least three generations to generate mutant *UQCRC1* heterozygous (HET) mice. Offspring generated from subsequent HET × HET intercrosses were genotyped via PCR via the primers VF1: 5′-CCACGTGGCCATTGCAGTAGA-3′ and VR1: 5′-TCGATACTCATGGCATCACAGAC-3′, followed by BsrGI digestion to detect the mutant allele [6]. Heterozygous mice were selected for experiments, with wild-type (WT) littermates used as controls or mitochondrial donors. Six-month-old animals, regardless of sex, were randomly assigned to experimental groups. All the mice were maintained under standard laboratory conditions with free access to food and tap water at the Laboratory Animal Center, Changhua Christian Hospital, Changhua, Taiwan.

While awake, the mice received a bilateral intranasal drop infusion of vehicle, 40 µg Mito, or CsA-binding Mito (Mito-CsA) in 10 µL MiR05 respiration buffer using a micropipette (QSP, 200 µL filter tips, Thermo Fisher Scientific, Waltham, MA, USA), as detailed previously [2]. This treatment regimen, which was administered once a week, was continued for a period of six months. The experiments were designed to minimize the number of animals used and their suffering.

### 2.4. Cell Culture and Mitochondrial Treatments

The wild-type (WT) and mutant *UQCRC1* knock-in human neuroblastoma SH-SY5Y cell lines were generously provided by the laboratory of Dr. Chin-Hsien Lin. They were individually transfected with CRISPR/Cas9 plasmids without site mutations (as a control, WT), or carried a UQCRC1 mutation (p.Y314S) heterozygous variant [5]. The confirmation of these cells was performed via PCR amplification and Sanger sequencing [6]. All the cell lines were cultured in a DMEM/F12 medium supplemented with 10% FBS, 100 U/mL penicillin, and 100 µg/mL streptomycin provided by Life Technologies (GIBCO BRL). The cells were maintained in a humidified atmosphere with 5% CO_2_. Given that isolated mitochondria can naturally enter human SH-SY5Y neuroblastoma cells [19], the effective dosage of mitochondria was determined on the basis of our previous studies [20]. A total of 100 μg of Mito or CsA-Mito, prepared as described above, was independently administered to 5 × 10^5^ host cells cultured in 10 cm dishes containing 10 mL of standard medium. The cells were then incubated for 24 h under standard culture conditions. Mitochondrial function assessments were performed after a 4-day recovery period.

### 2.5. Viability and Protruded Filopodia Assays

Cell viability was assessed via the WST-1 assay (Roche Diagnostics, Taipei, Taiwan). The morphology of the cells was examined via a bright-field Olympus BX43 microscope, whereas the lengths of the protrusions of filopodia were manually measured via ImageJ software (v1.54d, National Institutes of Health, Bethesda, MD, USA).

### 2.6. Mitochondrial Function

The mitochondrial fractions isolated from the mouse substantia nigra were obtained through homogenization via a class douncer. These homogenates were then used to evaluate CIII enzyme activity via a mitochondrial complex III activity assay (BioVision, Milpitas, CA, USA; Sigma Aldrich, St. Louis, MO, USA) following the manufacturer’s instructions. Additionally, mitochondrial respiration in saponin-permeabilized cells (1.25 ng/mL), involving basal respiration, ATP-linked production, and complex II/III activity, was assessed via high-resolution respirometry (Oxygraph-2k; Oroboros Instruments, Innsbruck, Austria) and quantified as respiratory oxygen flow (pmol/second/million cells). The analysis began by measuring basal respiration (BR) following the addition of cells. Basal respiration was defined as the recorded respiration in cells within MiR05 buffer without additional substrates or effectors. In contrast, ATP-linked production was defined as the reduction in respiration mediated by oligomycin (5 μM) in the presence of additional substrates, including glutamate (G, 10 mM), malate (M, 2 mM), and ADP (2.5 mM) (Sigma Aldrich, St Louis, MO, USA). Complex II/III activity was defined as the difference in respiration change between the addition of succinate (Suc) (10 mM) (Sigma Aldrich) and antimycin A (AA, 5 μM) (Sigma Aldrich).

### 2.7. Open-Field Analysis

To evaluate general motor behavior, an open-field test was conducted three months post-mitochondrial transplantation as previously described [2]. Briefly, the animals were placed in a 50 cm × 50 cm white plexiglass box following a 30-min adaptation period. Activity was recorded over two consecutive 15-min sessions via a ceiling-mounted video camera. Ethovision 3.1 software (Noldus, Leesburg, VA, USA) was used to measure distance, velocity, total zone boundaries crossed, and movement duration (seconds).

### 2.8. Striatal Dopamine Content with 6-[18F]-Fluoro-L-DOPA PET Analysis

The methodology for conducting 6-[18F]-fluoro-L-DOPA (FDOPA) PET scans in live animals was conducted in accordance with previously established procedures [21]. The mice were pretreated with carbidopa (25 mg/kg, i.v.) or entacapone (25 mg/kg, i.v.) 30 min prior to FDOPA injection (500 μCi, i.p.). Thirty minutes postinjection, the mice were anesthetized with isoflurane, and imaging was performed via a small animal PET/CT scanner (eXplore Vista DR, GE Healthcare, Fairfield, CT, USA).

### 2.9. Histological Analysis

Mice that underwent 6 months of treatment were euthanized via cardiac puncture under deep anesthesia induced by an overdose of inhaled isoflurane. Following euthanasia, intracardiac perfusion with saline was performed prior to tissue dissection. The brains were then extracted, fixed in 4% paraformaldehyde (Sigma-Aldrich) for 4 h, and cryoprotected in 30% (*w*/*v*) sucrose in PBS for 20 h. The brains were subsequently frozen, embedded in an Tissue-Tek OCT medium (Sakura Finetek USA, Torrance, CA, USA), and sectioned at a thickness of 5 or 10 μm. Alternatively, after fixation, the brains were dehydrated, cleared, and embedded in paraffin for histological sectioning.

For Nissl staining, glass-mounted sections were air-dried overnight before immersion in a 0.025% cresyl violet solution in 90 mM acetic acid (Sigma-Aldrich) and 10 mM sodium acetate (Sigma-Aldrich) for 3 h. Afterward, the slides were dehydrated in increasing ethanol and xylene concentrations before being cover-slipped with Histochoice mounting media (AMRESCO, Solon, OH, USA).

For immunohistochemical (IHC) staining, fixed sections were permeabilized and blocked for 30 min at room temperature. The samples were subsequently incubated overnight at 4 °C with a primary antibody, either anti-tyrosine hydroxylase (TH) (1:200 dilution, Novus Biologicals, Littleton, CO, USA) or anti-BrdU (1:200 dilution, Abcam, Cambridge, MA, USA), in a solution containing 1% BSA and 0.1% NaN3 (Sigma-Aldrich). After the samples were washed, they were incubated for 30 min at room temperature with an HRP-conjugated secondary antibody (1:200 dilution, Millipore, Billerica, MA, USA) in PBS. Following three washes in PBS, chromogenic detection of immunoreactivity was conducted via a DAKO 3,3′-diaminobenzidine (DAB) kit (DakoCytomation, Carpinteria, CA, USA) according to the manufacturer’s instructions. After chromogenic detection, a counterstain of Mayer’s hematoxylin (Sigma-Aldrich) was applied, and the slides were examined via light microscopy.

For fluorescence detection by costaining (IHC), a combination of primary antibodies from different species was used, followed by precomplexing of the mouse primary antibody BrdU with rabbit primary TH (1:400 dilution, Novus Biologicals), MBP (1:1000 dilution, Abcam), GFAP (1: 500 dilution, GeneTex, Hsinchu, Taiwan), and S100β (1:500 dilution, Abcam). The secondary antibodies used were Alexa Fluor™ 594-conjugated goat anti-rabbit (1:500 dilution, Jackson ImmunoResearch, West Grove, PA, USA) or DyLight 488-conjugated goat anti-mouse (1:500 dilution, Novus Biologicals), and the nuclei were visualized by counterstaining with 4′,6-diamidino-2-phenylindole (DAPI) (Abcam). Fluorescent signals were detected, and Z-stacks were acquired and analyzed via a confocal microscope (Olympus Fluoview FV1200, Olympus, Tokyo, Japan).

### 2.10. Mitochondrial Complex III Activity

The activity of mitochondrial complex III was assessed in fractionated mitochondrial samples derived from mouse substantia nigra homogenates. The assay was performed via a commercially available mitochondrial complex III activity kit (BioVision, Milpitas, CA, USA) following the methodology described by Kirby et al. [22].

### 2.11. Multiplex Cytokine Assay

After a six-month treatment period, the mouse blood collected via cardiac puncture was transferred into EDTA-containing Vacutainer tubes (Becton, Dickinson and Co., Franklin Lakes, NJ, USA). The extracted plasma was treated with a 1:100 dilution of proteinase inhibitor cocktail (Millipore #539134) and subsequently frozen at −80 °C for future testing. Multiple cytokines and chemokines in plasma were measured in duplicate via a Luminex platform (MAGPIX, Millipore, St. Charles, MO, USA) with a customized panel of MILLIPLEX MAP mouse Cytokine/Chemokine Magnetic Bead (MILLIPLEX MAP kits, EMD Millipore, Billerica, MA, USA) for the detection of interferon gamma (IFNγ), tumor necrosis factor-α (TNF-α), interleukin (IL)-1alpha (IL-1α), IL-1beta (IL-β), IL-6, IL-10, IL-12 (p70), CXCL1 (KC), and monocyte chemoattractant protein-1 (MCP-1). The data were analyzed with MILLIPLEX analyst software (v5.1, ViageneTech, Carlisle, MA, USA) according to the manufacturer’s instructions.

### 2.12. Statistical Analysis

Analyses were performed in triplicate for each experimental group. The biochemical data are presented as the means ± standard deviations, whereas the animal behavior test results are presented as the means ± standard errors of the means. Each group comprised six animals. Statistical associations were evaluated via a one-way ANOVA followed by Tukey’s post hoc *t*-tests for multiple comparisons. For comparisons involving paired samples, unpaired Student’s *t*-tests were used where appropriate. A *p*-value of less than 0.05 was considered statistically significant. Statistical analyses and graphics were performed via GraphPad Prism 5.0 software (GraphPad Software Inc., San Diego, CA, USA).

## 3. Results

### 3.1. Improvement of Locomotive Function in UQCRC1-Mutant Mice

To confirm the successful generation of *UQCRC1* p.Y314S knock-in (KI) mice, a CRISPR-Cas9 gene-editing strategy was used. As shown in Figure 1A (top), the targeted mutation (c.941A > C; p.Y314S) was introduced into exon 8, along with a BsrGI restriction site for allele-specific genotyping. PCR primers flanking the site amplified a 361 bp fragment from the WT allele, which was cut to 259 bp if the mutation was present. The genotyping results (Figure 1A, bottom) showed distinct patterns that wild-type mice had a 361 bp band, heterozygous KI mice had both 361 bp and 259 bp bands, and homozygous KI mice had only the 259 bp band. Compared with the dense trajectories of WT littermate controls, UQ mice, including those treated with the vehicle buffer (UQ-Sham), presented a marked reduction in locomotive activity, with sparse movement patterns. In contrast, both the UQ-Mito and UQ-Mito-CsA groups exhibited significant recovery, with movement patterns closely resembling those of the WT group (Figure 1B). Quantitative analysis of movement parameters, including total distance moved, duration, velocity, and zone crossings, consistently revealed that, compared with the UQ and UQ-Sham treatments, the UQ-Mito and UQ-Mito-CsA treatments led to substantial improvements. Although UQ-Mito showed slightly better levels of enhancement, the difference was not statistically significant (Figure 1C). These findings revealed that the transplantation of mitochondria pretreated with or without CsA significantly improved locomotion and exploratory behavior in UQ mice.

### 3.2. Recovery of Striatal ^18^F-DOPA Uptake, Survival of Tyrosine Hydroxylase-Positive Neurons, and Maintenance of Hippocampal Neuron Structure in the Treated Brain

To evaluate the impact of mitochondrial interventions on dopaminergic function, we conducted ^18^F-DOPA PET scans to assess the uptake of ^18^F-DOPA in the striatum (ST). The coronal and transverse axial PET images (left panel) showed the differences in ^18^F-DOPA uptake across the WT, UQ-Sham, UQ-Mito, and UQ-Mito-CsA groups. Compared with the WT controls, the UQ-Sham groups presented significantly reduced ^18^F-DOPA uptake, indicating impaired dopaminergic function. The quantification of the standardized uptake value (SUV) ratios (right panel, Figure 2A) relative to those of WT littermates revealed significant reductions in the UQ-Sham group bilaterally. Notably, both the UQ-Mito and UQ-Mito-CsA groups presented significant differences in the SUV ratios in both the right and left STs, with the UQ-Mito group showing the most significant increase but no clear difference from the UQ-Mito-CsA group.

The number of TH-positive neurons in the SN and ST, along with the morphology of hippocampal neurons, was assessed via IHC and Nissl staining. Compared with the WT controls, both the UQ and UQ-Sham groups presented a significant reduction in the number of TH-positive neurons in the SN and ST (Figure 2B, top and second rows). In contrast, the UQ-Mito and UQ-Mito-CsA groups demonstrated considerable recovery, with increased TH staining in both regions, nearing WT levels. To further evaluate hippocampal integrity, Nissl staining of the dentate gyrus (DG) was performed. Compared with the WT controls, the UQ and UQ-Sham groups presented a marked reduction in DG thickness and lower neuronal density in the granule cell layer (GCL) (Figure 2B, third and fourth rows). However, the UQ-Mito and UQ-Mito-CsA groups exhibited significant recovery of both DG thickness and neuronal density. Compared with the WT control group, the UQ and UQ-Sham groups presented notably reduced TH-positive neuron intensity in the SN and ST, as well as impaired DG thickness and GCL neuronal density, whereas the UQ-Mito and UQ-Mito-CsA groups presented marked improvements across all the parameters, which was consistent with the recovery observed in the WT controls (Figure 2C).

### 3.3. Restoration of Mitochondrial Activity in the Substantia Nigra of UQCRC1-Mutant Mice and Human Neuroblastoma Cells

Complex III (CIII) activity in the substantia nigra was significantly reduced in UQCRC1-mutant mice compared to the wild-type (WT) controls. Treatment with UQ-Mito or UQ-Mito-CsA significantly restored CIII activity toward WT levels (Figure 3A). Similarly, in UQCRC1-mutant neuroblastoma cells, mitochondrial respiration—including basal respiration (BR), ATP production linked to electron transport, and CIII-dependent respiration—was markedly impaired in the sham group relative to WT (Figure 3B, upper panel). Quantitative analysis (Figure 3B, lower panel) showed that both UQ-Mito and UQ-Mito-CsA treatments significantly improved mitochondrial respiratory function.

In UQCRC1-mutant cells, the sham group exhibited significantly reduced cell density and proliferation compared to the WT group (Figure 3C, upper and lower panels). These cells also showed impaired neurite outgrowth and shortened filopodia (Figure 3D, upper panel). Treatment with Mito or Mito-CsA significantly improved these parameters, restoring cell density, proliferation (Figure 3C), neurite outgrowth, and filopodia length (Figure 3D, lower panel) relative to the disease sham group. Importantly, no significant differences were observed between the two treatment groups across these measures (Figure 3A–D).

### 3.4. Reduction in Response to Inflammatory Cytokines and Chemokines

The multiplex analysis of cytokines and chemokines, as shown in Figure 4, revealed consistently higher levels in the UQ group than in the WT group, particularly for interleukin (IL)-1α, IL-6, IL-10, CXC chemokine ligand 1 (CXCL-1)/KC, and monocyte chemoattractant protein-1 (MCP-1)/chemokine ligand 2 (CCL2), with statistically significant differences observed. Sham treatment did not significantly alter these elevated levels. However, treatment with Mito or Mito-CsA resulted in a marked reduction in all the measured parameters, with the exception of a dramatic increase in IFN-γ. Notably, the Mito-CsA group exhibited a more pronounced inhibitory effect than the Mito group, although the difference was not statistically significant (Figure 4).

### 3.5. Distribution of Intranasally Administered Mitochondria Across Distinct Brain Regions

BrdU immunohistological staining was used to trace mitochondrial distribution six months post-intranasal administration. Figure 5A–E shows a brain atlas with highlighted regions and enlarged histological images. The brain regions highlighted in red individually indicate BrdU-positive signals in the ST, anterior commissure, substantia nigra pars compacta, cortex, hippocampal CA3, and lateral dorsal nucleus of the thalamus, where significantly more BrdU staining (deep brown color, as indicated by the arrow) was observed than in the sham group. Notably, mitochondrial expression was observed in the CA3 region of the hippocampus but not in the DG. This finding indicated a targeted and localized distribution of delivered mitochondria within specific brain regions. Moreover, no significant difference in mitochondrial distribution was observed between the Mito and Mito-CsA groups.

### 3.6. Exogenous Mitochondria Detected in Tyrosine Hydroxylase-Positive Neurons of the Substantia Nigra and Striatum, as Well as in Striatal Astrocytes, but Absent in Oligodendrocytes

Double immunohistological staining with BrdU and TH revealed that, in the sham group without transplantation, a small nonspecific BrdU signal appeared as light dots that gathered into slender aggregates in the SN (green fluorescence, indicated by arrow heads in Figure 6A). Sequential scanning of confocal Z-sections in the boxed area of the merged image confirmed the absence of co-expression with tyrosine hydroxylase (TH), a marker for DA neurons (red fluorescence) (Figure 6A, right panel). In contrast, the Mito transplantation group exhibited a prominent distribution of small, punctate BrdU signals (indicated by arrows) in the SN, which was markedly different from the nonspecific expression observed in the SN of the Sham group (Figure 6A) and was accompanied by TH-immunopositive expression shown by arrows in the boxed area of the merged image in Figure 6A. Furthermore, in the ST region, the Mito transplantation group presented punctate BrdU signals co-localized with TH-positive signals, in contrast to the sham group, which lacked both BrdU expression and false-positive signals (Figure 6B). To further quantify BrdU expression between the SN and ST, analysis excluding nonspecific BrdU expression in the SN revealed that BrdU expression in the ST was approximately 3.6 times higher than that in the SN (Figure 6C). Additionally, the proportion of TH signals co-expressing BrdU was approximately 5-fold greater in the ST than in the SN (47% vs. 8%, Figure 6D). To further quantify BrdU expression between the SN and ST, nonspecific BrdU expression in the SN was excluded from analysis, revealing that BrdU expression in the ST was approximately 3.6 times higher than that in the SN (Figure 6C). Moreover, the proportion of TH signals co-expressing BrdU was approximately 5-fold greater in the ST than in the SN (47% vs. 8%, Figure 6D). These findings indicate that TH-positive DA neurons incorporated BrdU-labeled mitochondria, with greater incorporation in the ST than in the SN.

To investigate whether the abundant BrdU signals in the ST region appear in other brain cell types, we analyzed merged and Z-section images with double staining for BrdU and myelin basic protein (MBP, an oligodendrocyte marker, Figure 6E) or glial fibrillary acidic protein (GFAP, an astrocyte marker, Figure 6F). The BrdU signals co-localized with GFAP (Figure 6F, arrows) but not with MBP, indicating that GFAP-labeled astrocytes incorporated the transplanted mitochondria.

### 3.7. Elevated Oligodendrocyte Population with Concurrent Reduction in Astrocyte Reactivity and Decreased S100β Expression in the Striatum

In addition, in the Mito group, MBP signals were significantly increased, whereas GFAP signals were suppressed compared with those in the Sham group (Figure 7A,B), indicating that mitochondrial transplantation increased the population of oligodendrocytes while simultaneously reducing that of astrocytes. Moreover, immunofluorescent staining of S100β, a marker for astrocytes, in the ST across different experimental groups revealed an increased S100β signal in the UQ and UQ-Sham groups compared to WT, indicating enhanced reactive ASTs in the disease mice. Notably, the UQ-Mito group exhibited a reduction in S100β expression relative to UQ-Sham, suggesting a potential modulatory effect of mitochondrial treatment on astrocyte activation (Figure 7C,D).

## 4. Discussion

The nasal transplantation of mitochondria has emerged as a promising therapeutic strategy for a range of neurological and systemic conditions, effectively bypassing the blood-brain barrier to enable direct mitochondrial transport to the central nervous system [2,23,24]. Our previous research demonstrated that in neurotoxin-induced Parkinson’s disease models, intranasal mitochondrial transplantation restored dopaminergic neuron function, reduced neuroinflammation, and improved locomotor activity [2]. Other studies highlight its potential in mitigating chemotherapy-induced cognitive deficits by enhancing cognition, reversing synaptic loss, and repairing myelin damage [25]. Additionally, functionalized intranasal mitochondrial transplantation has been shown to alleviate chemobrain and neuropathy in cisplatin-treated mice by modulating immune and neuronal pathways within the meninges [26]. In ischemic stroke models, intranasal mitochondrial administration has been shown to increase memory, upregulate synaptic marker expression, reduce oxidative stress, and restore mitochondrial function [27]. Collectively, these findings highlight the versatility and efficacy of nasal mitochondrial transplantation as a potential intervention for neurodegenerative and systemic disorders by targeting mitochondrial dysfunction. However, further studies are needed to determine whether variations in pathology within the same disease affect therapeutic outcomes and to elucidate the mechanisms governing mitochondrial distribution within the brain following nasal administration.

In light of the findings that mitochondria functionalized with dextran-triphenylphosphonium polymers can be nasally administered to mitigate neurotoxic effects [26]. We tend to utilize the properties of CsA, aiming for its natural accumulation in mitochondria due to its lipophilic nature, which enables passive diffusion across cellular membranes, including the outer and inner mitochondrial membranes. Furthermore, CsA binds to CypD, a mitochondrial matrix protein that regulates the mitochondrial mPTP, further promoting its mitochondrial localization. This study employed 5 μM CsA to determine its peak absorbance at 210 nm (Appendix A, upper panel). To quantify the degree of mitochondrial accumulation of CsA, the absorbance values were converted via a CsA standard curve (Appendix A, lower panel). At a CsA concentration of 5 μM, mitochondrial accumulation remained consistent across different mitochondrial concentrations (1, 3, and 5 μg/μL) (Appendix A). However, at 10 μg/μL, the intrinsic mitochondrial absorbance background exceeded that of CsA, and the CsA absorbance did not increase significantly, making quantification unreliable. Consequently, a mitochondrial concentration of 5 μg/μL was selected for CsA modification in subsequent experiments. Furthermore, experiments using cybrid cells harboring MERRF mitochondrial DNA mutations confirmed that CsA modification does not alter the ability of cells to internalize exogenous mitochondria across different doses (Appendix A). Unexpectedly, both in vitro and in vivo studies revealed that Mito-CsA did not enhance the therapeutic efficacy of mitochondrial transplantation by preventing mitochondrial damage, as anticipated. While Mito-CsA presented a greater suppression of disease-induced inflammatory cytokine expression compared to unmodified mitochondria. Therefore, we hypothesize that although in vitro studies demonstrated that CsA at this mitochondrial accumulation concentration effectively inhibited high calcium-induced mitochondrial swelling [16], the complexity of the in vivo pathological environment may account for its limited efficacy in enhancing mitochondrial transplantation outcomes. In addition to the mPTP pathway, additional mechanisms, such as oxidative stress, inflammation, and immune responses [28], may contribute to the attenuated therapeutic effect of Mito-CsA. Notably, we administered a single dose of CsA (10.31 ± 1.101 µg/mouse), equivalent to the CsA content accumulated by 40 µg of mitochondria, via intranasal administration at the same dosing frequency. This treatment did not ameliorate local motor function deficits (Appendix A) or enhance CIII activity within the SN (Appendix A). The findings indicate that even when released from Mito-CsA, this dose of CsA was insufficient to elicit neuroprotective effects. It is significantly lower than the therapeutic CsA concentrations for neuroprotection, which are typically reached through intraperitoneal administration at a dose of 1 mg/kg (equivalent to 25 µg for a 25 g adult mouse) [29].

Compared with our previous study [2], we observed therapeutic benefits of mitochondrial transplantation across distinct PD models. Weekly intranasal administration of mitochondria led to significant improvements in behavioral performance in both a neurotoxin-induced PD rat model (200 µg per rat for 3 months) [2] and a genetic PD mouse model harboring a *UQCRC1* gene mutation (40 µg per mouse for 6 months). Notably, the genetic PD model exhibited superior recovery, with behavioral outcomes approaching normal levels. This enhanced efficacy may be attributed to the relatively milder dopaminergic dysfunction observed in the genetic model (25–50% loss compared with 50–85% in the neurotoxin-induced model) and the early intervention prior to disease onset (treatment initiated at 6 months versus neurodegeneration occurring at 12 months in *UQCRC1* mutant mice [6]). On the other hand, distinct patterns of mitochondrial distribution were observed between the models. The extensive loss of SN dopaminergic neurons induced by a three-week medial forebrain bundle injection of the neurotoxin impaired the intracellular transport of transplanted mitochondria within the nigrostriatal pathway, resulting in their restricted localization to the ST [2]. In contrast, the genetic PD model, characterized by progressive neurodegeneration, permitted the detection of early-stage transplanted mitochondria in the SN, albeit at lower levels than in the ST. Thus, these differences indicate that the therapeutic efficacy of nasally delivered mitochondria for PD is influenced by variations in mitochondrial distribution within the brain, which are determined by the extent of cellular damage caused by different injury mechanisms.

Given the predominant localization of exogenous mitochondria in the ST, we examined their uptake by striatal OLs and ASTs. Only ASTs incorporated transplanted mitochondria, likely due to their high phagocytic and metabolic support capabilities, allowing them to rapidly take up exogenous mitochondria to meet energy demands [30]. In contrast, OLs, which are primarily responsible for myelin synthesis, exhibited minimal endocytic activity, as their function is focused on the production of myelin-associated lipids and proteins rather than the uptake of extracellular material [31]. Furthermore, the surface markers of exogenous mitochondria may have a greater affinity for AST receptors, facilitating their uptake, whereas OLs may lack the necessary binding sites for recognition and internalization [32]. Microenvironmental differences may also play a crucial role. Striatal ASTs may release signaling molecules that promote mitochondrial uptake, whereas OLs primarily rely on metabolic support from neighboring AST and neurons to meet their energy demands [33]. This finding implies that the localization and distribution of OLs may partially constrain their ability to directly take up exogenous mitochondria. Overall, the intranasal administration of exogenous mitochondria resulted in differential uptake among specific glial cell types, underscoring its importance for the development of targeted mitochondrial therapies for PD.

Mitochondrial transplantation increased OL population while attenuating reactive AST-related S100β expression, suggesting a neuroprotective effect, enhanced myelination, and improved neural conduction during disease progression. The upregulation of OLs suggests enhanced myelination or maintenance, contributing to the restoration of neural conduction [34]. Moreover, the reduction in reactive ASTs may indicate suppressed neuroinflammation, thereby further mitigating neural damage [30,35]. Mitochondrial targeted therapy has been shown to enhance energy metabolism, supporting neuronal and glial cell function recovery, as demonstrated in multiple sclerosis [35]. Maintaining cellular homeostasis is critical for neural stability, and studies have indicated that increasing the mitochondrial content in reactive ASTs mitigates metabolic stress (e.g., ATP deficiency), oxidative stress, and proinflammatory signaling, thereby reducing the demand for the astrocytic repair response and proliferation under pathological conditions such as neuroinflammation and neurodegeneration [36,37]. This finding aligns with our observation that compared with the sham control, intranasal mitochondrial therapy not only reduces astrocyte activation, but also consistently lowers plasma inflammatory cytokine levels. On the other hand, reactive ASTs are categorized into neurotoxic A1 type, which exacerbates neuronal damage, and neuroprotective A2 type, which supports neuronal survival, synaptic remodeling, and axonal regeneration [36]. Although this study did not specifically investigate the regulatory dynamics of A1 and A2 AST, the post-treatment reduction in S100β suggested a decrease in A1 ASTs, as its pro-inflammatory activity is associated with elevated S100β levels [38]. Additionally, evidence indicates that enhanced mitochondrial function and biogenesis can suppress proinflammatory A1 AST activation [37,39]. However, its effect on A2 astrocytes requires further investigation. Furthermore, the reduction in ASTs may also attenuate the formation of scar-forming ASTs, as indicated by a decrease in GFAP expression, a marker significantly upregulated during reactive astrogliosis and glial scar formation [40]. While GFAP activation plays a role in stabilizing injury sites, it may also create barriers to neural repair [41]. Given these dual effects, further investigation is needed to clarify how mitochondrion-targeted therapy influences astrocyte heterogeneity, and the balance between neuroprotection and repair.

## 5. Conclusions

This study validates that intranasal mitochondrial transplantation promotes neural repair and functional recovery in genetic models of PD. The therapeutic mechanism involves striatal oligodendrocyte populations and reduced astrocyte reactivity in ST, suggesting enhanced myelination and decreased neuroinflammation. Notably, exogenous mitochondrial incorporation into DA neurons was more prominent in the genetic PD model than in the neurotoxin-induced model, indicating that disease etiology affects mitochondrial distribution and treatment efficacy. Despite these differences, the intervention consistently demonstrated beneficial effects, underscoring its potential. Future efforts should focus on enhancing delivery precision and mitochondrial retention. If safety and technical challenges are resolved, intranasal mitochondrial therapy could represent a promising strategy for treating neurodegenerative diseases.

## Figures and Tables

**Figure 1 cells-14-01148-f001:**
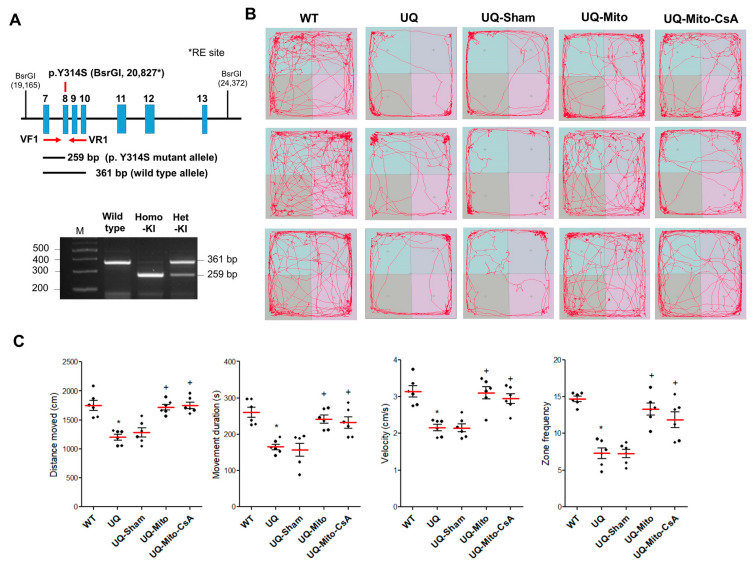
Genotype identification and locomotor behavior analysis in UQCRC1 (p.Y314S) knock-in (KI) mice. (**A**) CRISPR-Cas9-mediated targeting of the mouse *UQCRC1* gene and genotyping strategy. Top: Schematic representation of the mouse *UQCRC1* locus encompassing exons 7–13. The UQCRC1 p.Y314S KI (c.941A > C) mutation site was indicated in red within exon 8. The KI strategy incorporated an artificial BsrGI restriction enzyme (RE) site at position 20,827 to enable allele-specific genotyping. PCR primers VF1 and VR1 (red arrows) were designed to flank the mutation site and amplify a 361 bp fragment from the wild-type allele, and a 259 bp fragment from the mutant allele after BsrGI digestion. Bottom panel showed the genotyping of UQCRC1 KI mice by PCR and BsrGI restriction digestion. The wild-type allele yielded a 361 bp band, while the mutant KI allele produced a 259 bp fragment. The representative gel image displayed genotypes including wild-type, homozygous KI (Homo KI, +/+), heterozygous KI (Het KI, +/−), and a molecular weight marker (M, far left). (**B**) The behavioral performance of the wild-type, untreated and treated *UQCRC1* mutation KI mice were measured after 6 months of intranasal mitochondrial delivery (12 months of age). Representative track diagrams of open-field movement patterns for each group, illustrating locomotor activity differences. (**C**) Comparisons of the total distance traveled (cm), movement duration (s), velocity (cm/s), and zone transition frequency (the number of entries into specific areas) in the open field were performed among the groups (N = 6 per group). The one-way ANOVA revealed significant group differences in the distance traveled (F(4,25) = 17.79), movement duration (F(4,25) = 11.70), velocity (F(4,25) = 14.25), and zone transitions (F(4,25) = 21.19). Tukey’s post hoc tests indicated significant pairwise differences between WT and UQ (distance travel: *p* = 0.0002; movement duration: *p* = 0.0004; velocity: *p* = 0.0002; zone transitions: *p* < 0.0001) between UQ-Sham and UQ-Mito (distance travel: *p* = 0.0165; movement duration: *p* = 0.0018; velocity: *p* = 0.0003; zone transitions: *p* = 0.0001), and between UQ-Sham and UQ-Mito-CsA (distance travel: *p* = 0.0045; movement duration: *p* = 0.0057; velocity: *p* = 0.0023; zone transitions: *p* = 0.0018). The data are presented as the means ± SEMs. Statistical significance: *p* < 0.05, * indicates significant difference vs. WT; + indicates significant difference vs. UQ-Sham. Abbreviations: WT, untreated wild-type mice; UQ, untreated *UQCRC1* mutation knock-in mice; UQ-Sham, vehicle-treated disease mice; UQ-Mito, mitochondria-treated disease mice; UQ-CsA-Mito, CsA-accumulated mitochondria-treated disease mice.

**Figure 2 cells-14-01148-f002:**
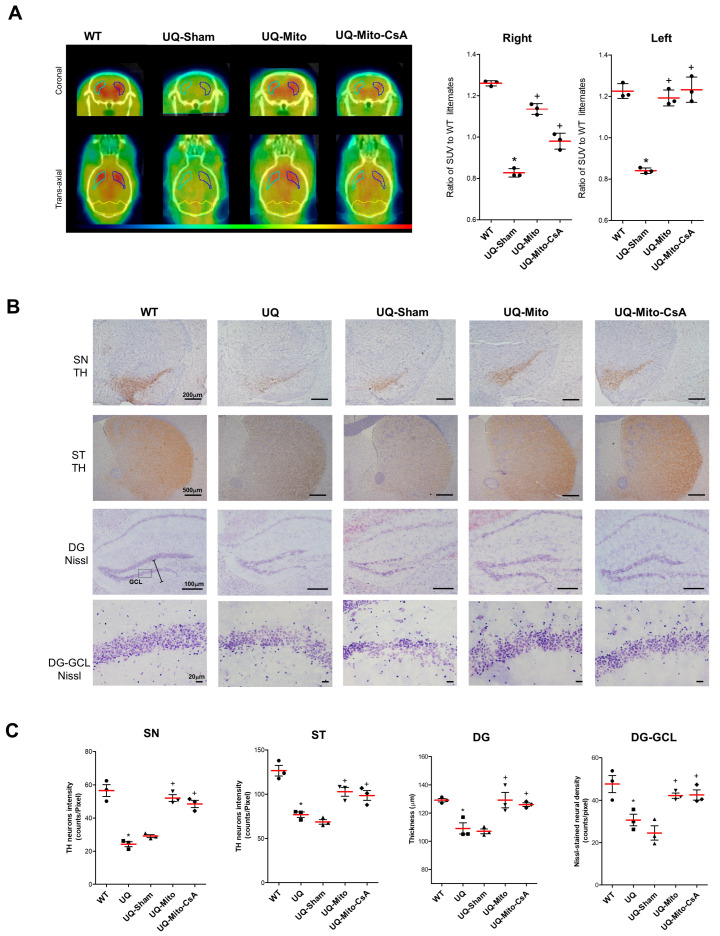
The levels of striatal dopamine, tyrosine hydroxylase (TH)-positive cells in the substantia nigra (SN) and striatum (ST), and hippocampal dentate gyrus (DG) neurons were evaluated in wild-type (WT) and UQCRC1 mutation (p.Y314S) knock-in mice after they received different treatments for 6 months. (**A**) Axial and coronal series of 6-[18F]-fluoro-L-DOPA PET images of 12-month-old wild-type and treated UQCRC1 mutation mice showing the bilateral striatum manually delineated to define the region of interest (ROI) on PET images. The color scale represented the relative uptake intensity, with higher uptake in red/yellow and lower uptake in blue, and the quantification of axial PET images is shown in the right panel. Dot plots show the ratio of SUV to WT littermates in the right and left hemispheres (N = 3 per group). A one-way ANOVA followed by Tukey’s post hoc test revealed significant group differences (Right: F(3,8) = 156.60; Left: F(3,8) = 63.94), with significant pairwise comparisons: WT vs. UQ-Sham (right and left: *p* < 0.0001), UQ-Sham vs. UQ-Mito (right and left: *p* < 0.0001), and UQ-Sham vs. UQ-Mito-CsA (right: *p* = 0.0004; left: *p* < 0.0001). (**B**) Representative immunohistochemical staining of the SN and ST illustrated the expression levels of TH-positive neurons, while Nissl staining of the hippocampal DG presented an overview of the DG structure, with the thickness marked by a scale bar resembling a ruler and the organization of the granule cell layer (GCL). The bottom row displays magnified views of the GCL, corresponding to the boxed regions from the images above, providing a more detailed examination of neuronal distribution and density. (**C**) TH-positive neurons in the substantia nigra (SN) and striatum (ST), along with dentate gyrus (DG) thickness and granule cell layer (GCL) neuronal density, were quantified. Data are presented as TH neuron intensity (counts/pixel) in the SN and in the ST, DG thickness (µm), and Nissl-stained neuronal density (counts/pixel) in the DG-GCL. Statistical analysis using a one-way ANOVA followed by Tukey’s post hoc test revealed significant group differences in the SN (F(4,10) = 42.09), ST (F(4,10) = 24.12), DG thickness (F(4,10) = 11.77), and DG-GCL neuronal density (F(4,10) = 10.68). Significant pairwise comparisons were observed between WT and UQ-Sham in all four regions (*p* < 0.0001 for SN and ST; *p* = 0.0048 for DG thickness; *p* = 0.0017 for DG-GCL), as well as between UQ-Sham and UQ-Mito (*p* = 0.0002 for SN; *p* = 0.0029 for ST; *p* = 0.0047 for DG thickness; *p* = 0.0112 for DG-GCL), and between UQ-Sham and UQ-Mito-CsA (*p* = 0.0008 for SN; *p* = 0.0073 for ST; *p* = 0.0133 for DG thickness; *p* = 0.0100 for DG-GCL). All data are presented as the means ± SEMs. Statistical significance: *p* < 0.05, ***** indicates significant difference vs. WT; + indicates significant difference vs. UQ-Sham. Abbreviations: WT, untreated wild-type mice; UQ, untreated UQCRC1 mutation knock-in mice; UQ-Sham, vehicle-treated disease mice; UQ-Mito, mitochondria-treated disease mice; UQ-CsA-Mito, CsA-accumulated mitochondria-treated disease mice.

**Figure 3 cells-14-01148-f003:**
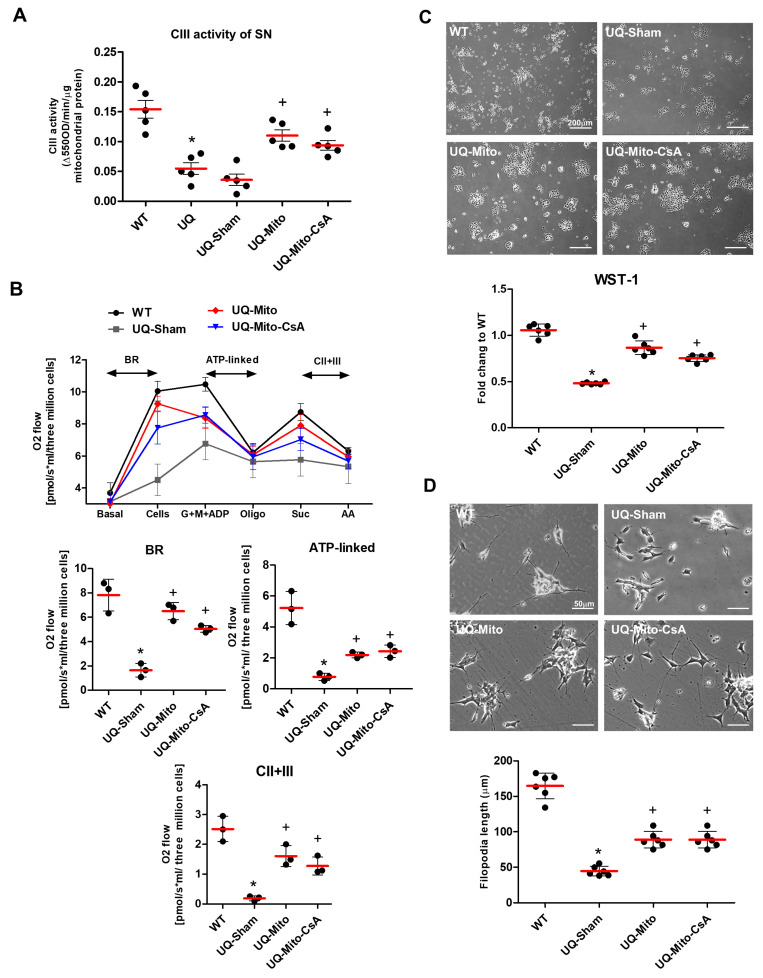
Mitochondrial function of the substantia nigra (SN) neurons of UQCRC1 mutation (p.Y314S) knock-in mice and neuroblastoma cells following mitochondrial treatments. (**A**) Mitochondrial complex III (CIII) activity in the substantia nigra was measured in wild-type and UQCRC1-mutant mice under different treatments (N = 5 per group). The one-way ANOVA showed significant group differences (F(4,20) = 9.99). Tukey’s post hoc test revealed reduced CIII activity in UQ vs. WT (*p* < 0.0001), and significant improvements in UQ-Mito (*p* = 0.0007) and UQ-Mito-CsA (*p* = 0.0081) compared to UQ-Sham. (**B**) Mitochondrial function in wild-type and UQCRC1 mutation knock-in SH-SY5Y neuroblastoma cells was analyzed following a 4-day recovery period after 24 h of mitochondrial delivery. Seahorse X-24 analysis of the oxygen consumption rate (OCR) was executed before and after the addition of cells across different respiratory states via different substrates and inhibitors of respiratory chain complexes (upper panel). The induced oxygen consumption (oxygen flux) was integrated and quantified via different approaches to provide an indirect measurement of mitochondrial activity (lower panel). These measurements included cell basal respiration (BR), ATP-linked respiration (assessed through oligomycin-mediated reduction), and CII + III-linked respiration (respiratory difference between succinate, a CII substrate, and antimycin A, a CIII inhibitor). The data are presented as the means ± SDs. N = 3 per group. The one-way ANOVA revealed significant group differences in BR (F(3,8) = 32.51), ATP-linked (F(3,8) = 30.13), and CII + III respiration (F(3,8) = 27.39). Tukey’s post hoc test showed significant impairment in UQ-Sham vs. WT (all *p* < 0.0001), with recovery in UQ-Mito (BR: *p* = 0.0004; ATP-linked: *p* = 0.0431; CII + III: *p* = 0.0027) and UQ-Mito-CsA (BR: *p* = 0.0038; ATP-linked: *p* = 0.0364; CII + III: *p* = 0.0133) compared to UQ-Sham. (**C**) Phase-contrast images (upper panel) depict cell morphology following a 4-day recovery period. Cell viability was assessed via the WST-1 assay (lower panel) and quantified as the fold change relative to the wild-type control. It revealed significant differences among experimental groups (F(3,20) = 122.96). Tukey’s post hoc test showed reduced activity in UQ-Sham compared to WT (*p* < 0.0001), while mitochondrial transfer significantly rescued activity in UQ-Mito (vs. UQ-Sham, *p* < 0.0001) and UQ-Mito-CsA (vs. UQ-Sham, *p* < 0.0001). The data are presented as the means ± SDs. N = 6 per group. (**D**) Phase-contrast images showing cell morphology and filopodium formation after a 4-day recovery period under different treatments. Filopodia length was quantified (lower panel) and presented as the mean ± SDs (μm). N = 6 per group. The one-way ANOVA revealed significant group differences (F(3,20) = 154.3). Tukey’s post hoc tests indicated significant differences between WT and UQ-Sham (*p* < 0.0001), UQ-Sham and UQ-Mito (*p* = 0.0021), and UQ-Sham and UQ-Mito-CsA (*p* = 0.0032). Statistical significance: *p* < 0.05, ***** indicates significant difference vs. WT; + indicates significant difference vs. UQ-Sham. Abbreviations: WT, untreated wild-type mice; UQ, untreated UQCRC1 mutation knock-in mice; UQ-Sham, vehicle-treated disease mice; UQ-Mito, mitochondria-treated disease mice; UQ-CsA-Mito, CsA-accumulated mitochondria-treated disease mice; BR, basal respiration; G, glutamate; M, malate; ADP, adenosine diphosphate; Oligo, oligomycin; Suc, succinate; AA, antimycin A.

**Figure 4 cells-14-01148-f004:**
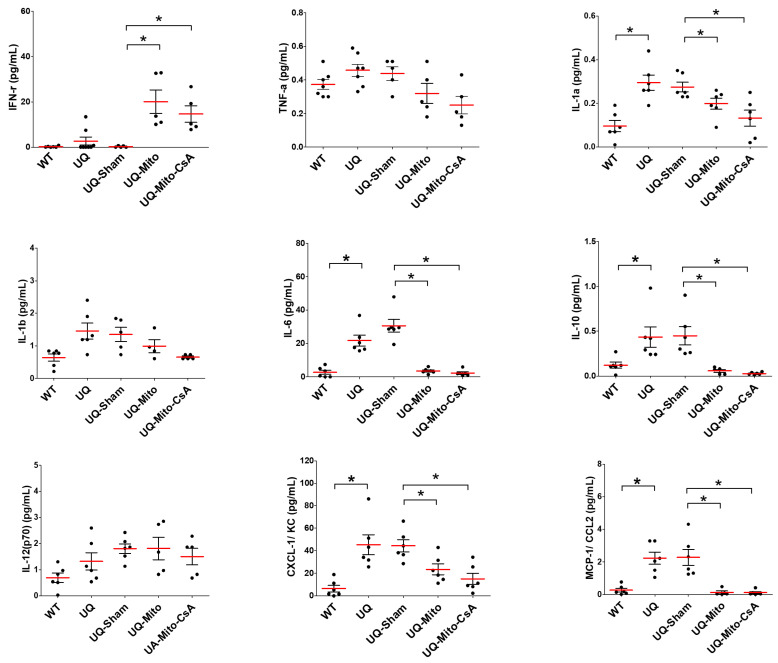
Levels of cytokine chemokines in plasma derived from wild-type mice and UQCRC1 mutation (p.Y314S) knock-in mice after receiving different treatments for 6 months. Data are presented as means ± SEMs (N = 5 per group at least). The one-way ANOVA revealed significant group differences for all cytokines analyzed. Tukey’s post hoc tests showed significant differences in IFN-γ [F(4,25) = 29.91; UQ-Sham vs. UQ-Mito (*p* = 0.0092), UQ-Mito-CsA vs. UQ-Sham (*p* = 0.0054)], IL-1α [F(4,25) = 8.68; WT vs. UQ (*p* = 0.0031), UQ-Sham vs. UQ-Mito (*p* = 0.0459), UQ-Sham vs. UQ-Mito-CsA (*p* = 0.0209)], IL-6 [F(4,25) = 31.16; WT vs. UQ (*p* < 0.0001), UQ-Sham vs. UQ-Mito (*p* < 0.0001), UQ-Sham vs. UQ-Mito-CsA (*p* < 0.0001)], IL-10 [F(4,25) = 7.99; WT vs. UQ (*p* = 0.027), UQ-Sham vs. UQ-Mito (*p* = 0.003), UQ-Sham vs. UQ-Mito-CsA (*p* = 0.001)], CXCL-1/KC [F(4,25) = 9.08; WT vs. UQ (*p* = 0.0006), UQ-Sham vs. UQ-Mito (*p* = 0.0136), UQ-Sham vs. UQ-Mito-CsA (*p* = 0.0137)], and MCP-1/CCL2 [F(4,25) = 16.30; WT vs. UQ (*p* < 0.0001), UQ-Sham vs. UQ-Mito (*p* < 0.0001), UQ-Sham vs. UQ-Mito-CsA (*p* = 0.0003)]. Statistical significance: *p* < 0.05, * indicates significant difference between groups. Abbreviations: WT, untreated wild-type mice; UQ, untreated UQCRC1 mutation knock-in mice; UQ-Sham, vehicle-treated disease mice; UQ-Mito, mitochondria-treated disease mice; UQ-CsA-Mito, CsA-accumulated mitochondria-treated disease mice; IFN-γ, interferon-gamma; TNF-α, tumor necrosis factor-alpha; IL-1α, interleukin-1 alpha; IL-1β, interleukin-1 beta; IL-6, interleukin-6; IL-10, interleukin-10; IL-12(p70), interleukin-12 p70; CXCL-1/KC, CXC chemokine ligand 1/keratinocyte/chemoattractant; MCP-1/CCL2, monocyte chemoattractant protein-1/chemokine ligand 2.

**Figure 5 cells-14-01148-f005:**
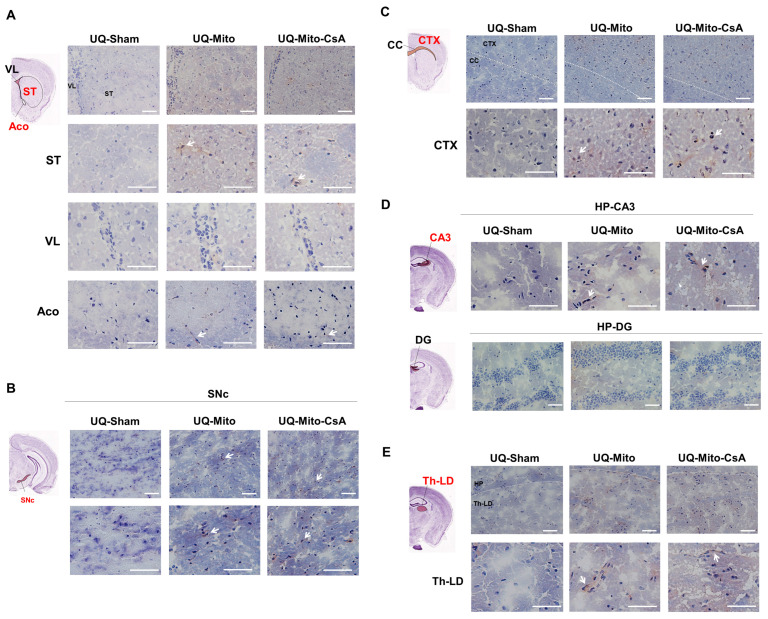
Differences in the brain distribution of delivered mitochondria with or without CsA treatment in UQCRC1 mutation (p.Y314S) knock-in mice. BrdU immunohistological staining was used to trace the mitochondrial distribution at 6 months post-intranasal administration. (**A**–**E**) shows the brain atlases of different regions with highlighted areas and corresponding enlarged histological images. The brain regions highlighted in red indicate BrdU-positive signals in the (**A**) striatum (ST) and anterior commissure (Aco), (**B**) substantia nigra pars compacta (SNc), (**C**) cortex (CTX), (**D**) hippocampal CA3, and (**E**) lateral dorsal nucleus of the thalamus (Th-LD), where significantly more BrdU staining (deep brown color, as indicated by the arrow) was observed than in the sham group. N = 3 per group. Scale bar: 50 μm. Abbreviations: UQ-Sham, vehicle-treated diseased mice; UQ-Mito, mitochondria-treated diseased mice; UQ-CsA-Mito, CsA-accumulated mitochondria-treated diseased mice; LV, lateral ventricle; CC, corpus callosum.

**Figure 6 cells-14-01148-f006:**
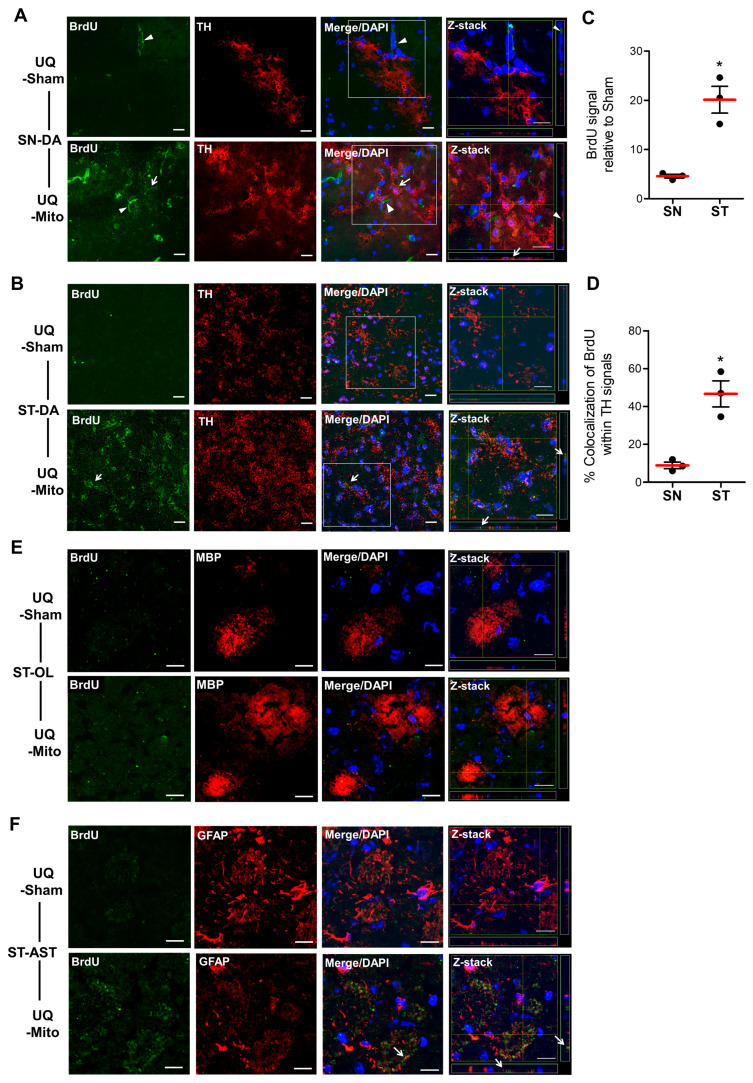
Internalization of BrdU-labeled mitochondria via intranasal delivery in dopaminergic (DA) neurons of the substantia nigra pars (SN) and striatum, (ST) and in oligodendrocytes (OL) and astrocytes (AST) of the ST in UQCRC1 mutation (p.Y314S) knock-in mice. (**A**) Double immunohistological staining with BrdU (green fluorescence) and tyrosine hydroxylase (TH, red fluorescence) was performed to evaluate the uptake of delivered mitochondria in SN DA neurons. The nonspecific BrdU signals in the sham control group, which appeared as light green dots forming slender aggregates, were indicated in the SN (arrowheads). Confocal Z-section analysis of the boxed area confirmed the absence of BrdU and TH co-localization (right panel). In contrast, a prominent distribution of punctate BrdU signals was observed (arrows) in the mitochondria-treated SN, with co-localization of BrdU with TH-positive signals (arrows in the boxed area) shown in the corresponding merged Z-sectioned image. (**B**) In the ST region, the Mito transplantation group presented BrdU signals co-localized with TH-positive neurons (arrows), whereas the sham group lacked BrdU expression and displayed no false-positive signals. (**C**) Quantification of BrdU expression relative to the sham group, excluding nonspecific signals. An unpaired t-test revealed a significant increase in BrdU expression in ST compared to the SN (t = 5.63, *p* = 0.0049). (**D**) The proportions of TH-positive neurons co-expressing BrdU in the SN and ST were quantified. An unpaired t-test revealed a significant increase in the ST compared to the SN (t = 5.32, *p* = 0.0258). All data are presented as the means ± SEMs (N = 3 per group). (**E**) BrdU distribution and co-localization of BrdU-labeled mitochondria with myelin basic protein (MBP)-positive oligodendrocytes (OLs) and (**F**) glial fibrillary acidic protein (GFAP)-positive astrocytes (ASTs) were examined as outlined earlier in the ST of UQCRC1 mutation knock-in mice. Representative immunofluorescence images showing BrdU (green fluorescence), cell-specific markers (red fluorescence), and DAPI nuclear staining (blue fluorescence) in the sham and mitochondria-treated groups. In the mitochondria-treated groups, BrdU was undetectable in OLs, whereas ASTs displayed BrdU expression. Confocal Z-section analysis of the merged images confirmed mitochondrial localization (arrows) within striatal ASTs (right panel). The data are presented as the means ± SEMs, with statistical significance indicated by asterisks (*). N = 3 per group. Scale bars: 20 µm. Abbreviations: UQ-Sham, vehicle-treated diseased mice; UQ-Mito, mitochondria-treated diseased mice.

**Figure 7 cells-14-01148-f007:**
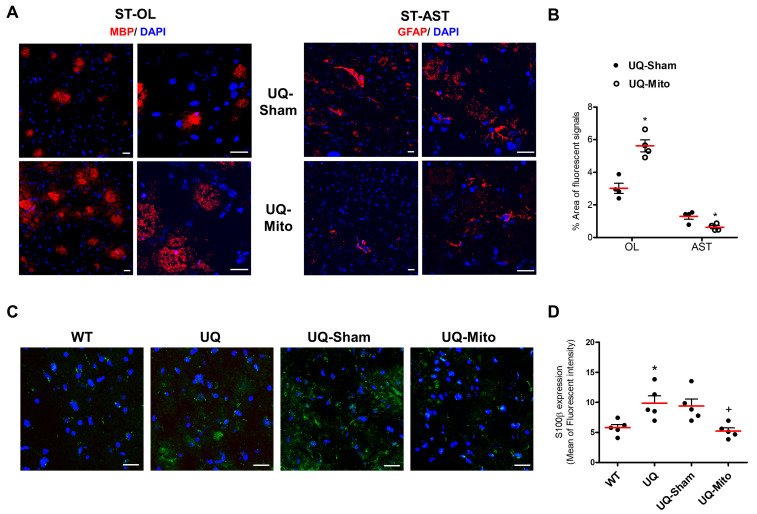
Comparison of the striatal neuron populations of oligodendrocytes (OLs), astrocytes (ASTs), and reactive astrocytes (ASTs). (**A**) Immunohistochemistry (IHC) analysis using cell-specific markers (red fluorescence) and DAPI nuclear staining (blue fluorescence) showing myelin basic protein (MBP)-positive OLs and glial fibrillary acidic protein (GFAP)-positive astrocytes ASTs in the striatum (ST) in the mitochondria-treated and sham groups. (**B**) Quantification of fluorescence-positive areas for OLs and ASTs was performed using ImageJ software, measuring the relative area of red fluorescence signals. Unpaired *t*-tests showed increased OL (t = 4.94, *p* = 0.0041) and decreased ASTs (t = 3.46, *p* = 0.018) in the UQ-Mito group vs. UQ-Sham. Data are the means ± SEMs (N = 4 per group). (**C**) Expression of striatal AST-positive neurons labeled by S100β IHC in wild-type and UQCRC1 mutation knock-in mice under different treatment conditions. (**D**) Quantification and comparison of S100β expression were performed between groups. A one-way ANOVA followed by Tukey’s post hoc test revealed significant differences [F(3,16) = 4.18] in expression levels between WT and UQ (*p* = 0.0487) and between UQ-Sham and UQ-Mito (*p* = 0.0050). Statistical significance: *p* < 0.05, * indicates significant difference vs. WT; + indicates significant difference vs. UQ-Sham. Data are presented as the means ± SEMs (N = 5 per group). Scale bars: 20 µm. Abbreviations: WT, untreated wild-type mice; UQ, untreated UQCRC1 mutation knock-in mice; UQ-Sham, vehicle-treated disease mice; UQ-Mito, mitochondria-treated disease mice.

## Data Availability

The data are available from the corresponding author upon reasonable request.

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
