# Peer review of "Intranasal Mitochondrial Transplantation Restores Mitochondrial Function and Modulates Glial–Neuronal Interactions in a Genetic Parkinson’s Disease Model of *UQCRC1* Mutation"

_cells, 2025, doi:10.3390/cells14151148_

Round 1

Reviewer 1 Report

Comments and Suggestions for Authors The manuscript by Chin-San Liu et al., entitled "Intranasal Mitochondrial Transplantation Restores Mitochondrial Function and Modulates Glial-Neuronal Interactions in a genetic Parkinson's Disease Model of UQCRC1 mutation" describes about the application of non-invasive mitochondrial transplant reduces the Parkinson's symptoms. The manuscript is well written. Incorporating some minor changes mentioned below will further improve the quality of the manuscript:   1. The authors forgot to add Figure-3. Kindly add the Figure-3 in the main manuscript.   2. Line 478, " Authors Nasal transplantation of mitochondria ....." Kindly remove the word "Authors".   3. In Line 469, Figure 7B, the quantification highlights differences in expression; however, the authors did not provide information regarding the statistical methods or imaging software used for quantification. Kindly include these details.   4. I would recommend measuring additional components of the mitochondrial electron transport chain, such as Complexes II, and IV across all different experimental groups. 

Author Response

Reviewer 1

Comments 1. The authors forgot to add Figure-3. Kindly add the Figure-3 in the main manuscript. 

Response 1: Thank you for your correction. We have added Figure 3 with legend as suggested in the revised manuscript (Line 381-417, highlighted in red).

Comments 2. Line 478, " Authors Nasal transplantation of mitochondria ....." Kindly remove the word "Authors".   

Response 2:  The modified version has removed the Typing error. Thank you for your careful correction.

Comments 3. In Line 469, Figure 7B, the quantification highlights differences in expression; however, the authors did not provide information regarding the statistical methods or imaging software used for quantification. Kindly include these details.

Response 3:  Thank you for pointing this out. The quantification shown in Figure 7B was performed using ImageJ software to measure the fluorescent signal–positive area for MBP- and GFAP-stained cells in the striatum. Statistical analysis was conducted using unpair unpaired Student’s t-test. These details have been added to the revised Figure 7B legend (Line 553-556, highlighted in red).

Comments 4. I would recommend measuring additional components of the mitochondrial electron transport chain, such as Complexes II, and IV across all different experimental groups. 

Response 4: Previous studies have shown that UQCRC1 mutant cells specifically affect Complex III activity without impairing Complex II or Complex IV function (as shown in the figure below, with reference link provided). Therefore, in this study, we focused on analyzing Complex III activity using an ELISA kit, and assessed the combined function of Complex II + III using O2k respirometry.

That said, we fully agree with your suggestion that measuring the activity of individual ETC complexes would improve the overall quality and robustness of the study. We will incorporate this approach in our future experiments. Thank you for your valuable suggestion.

Figure (F–H) The individual activity of mitochondrial respiratory chain complex I–IV was examined further in wild-type cells and UQCRC1 p.Tyr314Ser knock-in SH-SY5Y cells. Representative seahorse profiles for the individual activity of mitochondrial respiratory chain complexes I–IV are illustrated in wild-type cells (F) and UQCRC1 p.Tyr314Ser knock-in SH-SY5Y cells (G) by measuring the OCRs before (basal respiration) and after sequential additions of oligomycin (ATP synthase inhibitor), CCCP (uncoupling protonophore), and then rotenone (complex I inhibitor) and succinate (complex II inhibitor) (blue lines); duroquinol (complex III inhibitor) (red lines); and TMPD and ascorbate (complex IV inhibitors) (green lines). This strategy allowed for the determination of the contribution of each component of respiration chain complexes. (H) The percentages of OCR attributable to the activities of complexes I–IV.

https://pubmed.ncbi.nlm.nih.gov/33141179/

Reviewer 2 Report

Comments and Suggestions for Authors

 Dear authors,

The authors reported intranasal mitochondrial transplantation is associated with Parkinson's disease through glial-neuronal interactions. I believe many readers will be interested in the topics and quality of the study is relatively high. Since it is not suitable for publication in the journal now, I think revision is needed. Reviewer's comments as below.

(1) It's very hard to find BrdU-positive signal in Figure 5. Authors have to show high-magnificent images and indicate all BrdU-positive signal.

(2) Does the treatment affect oligodendrocyte progenitor cell's proliferation? Only myelination?

(3) Does the number of synapse in dopaminergic neurons, that connect with spiny neurons (GABAnergic neurons), in striatum? 

Author Response

Comments 1: It's very hard to find BrdU-positive signal in Figure 5. Authors have to show high-magnificent images and indicate all BrdU-positive signal.

Response 1: As requested, we have revised Figure 5 to include high-magnification images clearly indicating all BrdU-positive signals.

Comments 2: Does the treatment affect oligodendrocyte progenitor cell's proliferation? Only myelination?

Response 2: This is an excellent question. In this study, we focused specifically on the analysis of myelination, using Myelin Basic Protein as a marker for myelin components. However, we did not assess oligodendrocyte proliferation using Ki-67. Based on our findings, the transplanted mitochondria did not enter oligodendrocytes, suggesting that the mitochondrial effects are unlikely to directly regulate oligodendrocyte progenitor cell activity. Furthermore, myelination is a result of oligodendrocyte differentiation, rather than their proliferation. Therefore, we speculate that mitochondrial treatment may indirectly influence oligodendrocyte differentiation, rather than promoting cell proliferation.

Comments 3. Does the number of synapse in dopaminergic neurons, that connect with spiny neurons (GABAnergic neurons), in striatum?

Response 3: Thank you for this insightful comment. Although dopaminergic neurons are not the primary source of excitatory input to the striatum, they form functional modulatory connections with GABAergic medium spiny neurons (MSNs), primarily through dopamine receptor signaling. Restoration of dopaminergic function does not directly enhance excitatory synaptic transmission, but rather modulates glutamatergic inputs from the cortex and thalamus via regulation of D1R- and D2R-expressing MSNs. This modulatory effect helps re-establish the balance between the direct and indirect basal ganglia pathways, thereby improving striatal output and motor function. While our current study did not directly investigate this mechanism, we agree that it is an important and intriguing direction for future research.

Reviewer 3 Report

Comments and Suggestions for Authors

The manuscript titled "Intranasal Mitochondrial Transplantation Restores Mitochondrial Function and Modulates Glial-Neuronal Interactions in a Genetic Parkinson's Disease Model of UQCRC1 Mutation" presents a novel and promising therapeutic approach for Parkinson’s disease (PD). The study explores the efficacy of intranasal delivery of exogenous mitochondria, a non-invasive strategy, in restoring mitochondrial function and mitigating neurodegenerative processes in a genetic PD model harboring a UQCRC1 mutation. The authors employ a comprehensive array of experimental techniques to demonstrate the neuroprotective benefits of mitochondrial transplantation.

Major Comments:

  • In the Materials and Methods section (line 100), the authors describe mitochondrial labeling; however, the manuscript would benefit from the inclusion of co-localization images showing BrdU with MitoTracker Red. Such images would provide visual confirmation of BrdU localization within mitochondria and support the specificity of mitochondrial labeling.
  • The authors are encouraged to evaluate mitochondrial quality following BrdU labeling by assessing mitochondrial membrane potential and respiratory activity. These measurements would help determine whether the labeling process affects mitochondrial integrity or function.
  • In Figure 1, the authors should include the protein expression analysis of UQCRC1 in UQCRC1 knockout (KO) mice to validate the KO model at the molecular level.
  • Although the Materials and Methods section (line 141) mentions the use of UQCRC1 knock-in human neuroblastoma SH-SY5Y cell lines, corresponding protein expression data for UQCRC1 are not provided. Inclusion of these results is important to confirm the genetic manipulation.
  • The authors refer to mitochondrial function assessments in the Methods (line 161), but the corresponding results are absent from the manuscript. Please include the data to substantiate this aspect of the study.
  • Line 333 mentions Complex III (CIII) activity and references Figure 3, but this figure appears to be missing from the manuscript. Please ensure that all referenced figures are included and correctly labeled.
  • In Figure 6, the BrdU images are unclear across multiple panels. Higher-resolution images or improved contrast would enhance interpretability and ensure the clarity of mitochondrial labeling and localization data.

Author Response

Comments 1. In the Materials and Methods section (line 100), the authors describe mitochondrial labeling; however, the manuscript would benefit from the inclusion of co-localization images showing BrdU with MitoTracker Red. Such images would provide visual confirmation of BrdU localization within mitochondria and support the specificity of mitochondrial labeling. The authors are encouraged to evaluate mitochondrial quality following BrdU labeling by assessing mitochondrial membrane potential and respiratory activity. These measurements would help determine whether the labeling process affects mitochondrial integrity or function.

Response 1: We fully agree with your suggestions. In our earlier study of mitochondrial transplantation, we validated BrdU as a reliable label for mitochondria by co-localization with MitoTracker (see figure below, with reference link provided). Flow cytometric analysis of isolated mitochondria (1 mg/mL) co-stained with anti-BrdU–Alexa Fluor 647 (32.5 μM) and MitoTracker Green (500 nM) demonstrated that over 95 % of mitochondria were MitoTracker-positive (Q1, top left), while 49.9 ± 2.05 % co-expressed BrdU (Q2, top right). Under these conditions, the effective binding affinity of the BrdU antibody was 15.3 μg mitochondria per μM BrdU. These details have been added to the revised manuscript (Materials and Methods, Line 103-104; adding Ref. 17; highlighted in red).

Figure (B) The antibody binding affinity of BrdU was analyzed by co-staining isolated mitochondria (1 mg/mL) with BrdU antibody tagged with Alexa Fluor 647 dye (32.5 μM) and MitoTracker Green (500 nM), a mitochondria-specific dye, using flow cytometry.

https://www.dovepress.com/antitumor-actions-of-intratumoral-delivery-of-membrane-fused-mitochond-peer-reviewed-fulltext-article-OTT

Furthermore, our in vivo BrdU labeling protocol was adapted from Battersby & Shoubridge (HMG 2001, https://academic.oup.com/hmg/article/10/22/2469/670265), in which 1 mg/g body weight was used to track mtDNA replication without evaluation of mitochondrial function. In our studies, we halved this dose and confirmed the treatment viability of BrdU-labeled mitochondria across multiple animal models. Nevertheless, we appreciate your concern and recognize the importance of verifying that BrdU incorporation does not compromise mitochondrial integrity. Therefore, in future work we will directly assess mitochondrial membrane potential and respiratory activity following BrdU labeling.

Comments 2. In Figure 1, the authors should include the protein expression analysis of UQCRC1 in UQCRC1 knockout (KO) mice to validate the KO model at the molecular level. Although the Materials and Methods section (line 141) mentions the use of UQCRC1 knock-in human neuroblastoma SH-SY5Y cell lines, corresponding protein expression data for UQCRC1 are not provided. Inclusion of these results is important to confirm the genetic manipulation.

Response 2: Thank you for your valuable comment, and we apologize for the unclear description that may have caused misunderstanding. As suggested, we have included a schematic diagram of the knock-in targeting strategy for the mutant UQCRC1 allele in mice, along with the corresponding genotyping results, in the revised Figure 1 and its legend (Line 299-310, highlighted in red). The detailed protocol for generating the knock-in allele—including full sequence information and molecular validation procedures—has been described in our previous publication, which is appropriately cited in the Materials and Methods section (Ref. 6).

Comments 3. The authors refer to mitochondrial function assessments in the Methods (line 161), but the corresponding results are absent from the manuscript. Please include the data to substantiate this aspect of the study.

Response 3: The results of the mitochondrial function assessments are presented in Figure 3. We apologize for the oversight and have now inserted Figure 3 and legend into the revised manuscript (Line 381-417, highlighted in red).   

Comments 4. Line 333 mentions Complex III (CIII) activity and references Figure 3, but this figure appears to be missing from the manuscript. Please ensure that all referenced figures are included and correctly labeled.

Response 4: Thank you for your correction. We have added Figure 3 and legend including the performance of CIII activity of SN (Figure 3A) as suggested in the revised manuscript (Line 381-417, highlighted in red).

Comments 5. In Figure 6, the BrdU images are unclear across multiple panels. Higher-resolution images or improved contrast would enhance interpretability and ensure the clarity of mitochondrial labeling and localization data.

Response 5: We thank the reviewer for the suggestion. In response, we have revised Figure 6 by replacing the BrdU panels with higher-resolution images, enhancing contrast to improve clarity, and enlarging the key regions to better show BrdU expression and its overlap with different neuronal markers. These adjustments improve the visibility and interpretability of mitochondrial labeling and colocalization. The updated Figure 6 has been incorporated into the revised manuscript.

Reviewer 4 Report

Comments and Suggestions for Authors

The manuscript “Intranasal Mitochondrial Transplantation Restores Mitochondrial Function and Modulates Glial-Neuronal Interactions in a genetic Parkinson's Disease Model of UQCRC1 mutation” by Chang is a research article which validated transplantation of mitochondria with or without Cyclosporin A (CsA) preloading as a method to treat mitochondrial dysfunction and improve disease progression through intranasal delivery. Liver‐derived mitochondria were labeled with bromodeoxyuridine (BrdU), incubated with CsA to inhibit mPTP opening or not, and administered weekly via the nasal route to 6-month-old mice for six months. The authors found that both treatment groups showed significant locomotor improvements in open-field tests and that PET imaging showed increased striatal tracer uptake, indicating enhanced dopamine synthesis capacity. In addition, the immunohistochemical analysis showed increased neuron survival in the dentate gyrus, a higher number of tyrosine hydroxylase (TH)-positive neurons in the substantia nigra (SN) and striatum (ST), and a thicker granule cell layer. Thus, intranasally mitochondrial transplantation shows neuroprotective effects in genetic PD, validating a noninvasive therapeutic approach. In general, this article is critical in this field and contains essential contents. I have minor concerns before this manuscript is accepted for publication.

The authors used t-test and one-way ANOVA in this manuscript. Please add t and F values in the text or figure legends.

In bar graphs, all the data plots should be added because the readers can obtain useful information from these data.

Author Response

Comments 1. The authors used t-test and one-way ANOVA in this manuscript. Please add t and F values in the text or figure legends.

Response 1: Thank you for your valuable comment. We have now added the corresponding F-values and t-values in the figure legends where statistical analyses were performed using One-way ANOVA followed by Tukey’s post hoc test or unpaired t-tests.

Comments 2. In bar graphs, all the data plots should be added because the readers can obtain useful information from these data.

Response 2: In response, we have replaced all bar graphs with dot plots to display individual data points. Thank you for your helpful suggestion.

Round 2

Reviewer 3 Report

Comments and Suggestions for Authors

-